# Homophobic Bullying Among Adolescents: Prevalence, Associations with Emotional Factors, Psychopathological Symptoms, and Predictors

**DOI:** 10.3390/healthcare13233119

**Published:** 2025-12-01

**Authors:** Maite Garaigordobil, Juan Pablo Mollo-Torrico, Mónica Rodríguez-Enríquez

**Affiliations:** 1Department of Personality, Assessment, and Psychological Treatments, University of the Basque Country, 20018 San Sebastián, Spain; juanpablomollot@icloud.com; 2Department of Developmental Psychology and Communication, University of Vigo, 32002 Ourense, Spain; monica.rodriguez.enriquez@uvigo.gal

**Keywords:** homophobic bullying, LGBTQI+, prevalence, sexual orientation, emotional development, psychopathology, predictors

## Abstract

**Background/Objectives**: Despite progress in recognizing sexual diversity, homophobic bullying persists. This study had four objectives: (1) to identify the prevalence of homophobic bullying (victims, perpetrators, and bystanders); (2) to explore whether differences exist between victims and perpetrators as a function of sexual orientation with respect to emotional factors and psychopathological symptoms; (3) to analyze whether victims and perpetrators of homophobic bullying have sought psychological assistance significantly more often; and (4) to identify predictive variables of victimization and perpetration of homophobic bullying. **Methods**: The sample comprised 1558 Bolivian students aged 13 to 17 years (M = 14.64; SD = 0.96), who completed six standardized assessment instruments. **Results**: (1) A substantial percentage of students reported homophobic bullying behaviors. Victims: 76.6% reported experiencing homophobic behaviors, with significantly higher rates among non-heterosexual students (χ^2^ = 7.40, *p* < 0.01) and no gender differences (χ^2^ = 0.013, *p* > 0.05). Perpetrators: 11.8% admitted engaging in homophobic aggressive behaviors, with no differences by sexual orientation (χ^2^ = 0.306, *p* > 0.05) but significantly higher rates among males (χ^2^ = 8.49, *p* < 0.01). Bystanders: 51.9% reported witnessing homophobic behaviors, with significantly higher prevalence among non-heterosexual students (χ^2^ = 7.03, *p* < 0.01) and females (χ^2^ = 4.98, *p* < 0.05). (2) Analyses of variance showed that non-heterosexual victims scored significantly lower on emotional regulation, empathic joy, overall empathy, and happiness, and significantly higher on fear of negative social evaluation, overall social anxiety, all psychopathological symptom dimensions assessed (somatization, obsession–compulsion, interpersonal sensitivity, depression, anxiety, hostility, phobic anxiety, paranoid ideation, psychoticism), and the global severity index. Non-heterosexual perpetrators also displayed significantly higher scores on several psychopathological symptoms (depression, anxiety, hostility, paranoid ideation, psychoticism) and on the global severity index. Effect sizes were moderate for psychopathological symptoms and small for emotional variables. (3) Victims (OR = 1.392, 95% CI [1.04, 1.86], *p* = 0.024) and perpetrators (OR = 1.507, 95% CI [1.07, 2.10], *p* = 0.017) of homophobic bullying reported significantly higher rates of seeking psychological assistance in the past year compared to those uninvolved in bullying. (4) Hierarchical regression analyses identified significant predictors of victimization (R^2^ = 18.6%): non-heterosexual orientation, male gender, higher somatization, paranoid ideation, fear of negative evaluation, and lower happiness. For perpetration, only being male and higher levels of phobic anxiety emerged as significant predictors in the final model, explaining 5.1% of the variance. **Conclusions**: The findings underscore the urgency of implementing school-based psychoeducational anti-bullying prevention programs that include activities designed to foster tolerance toward sexual diversity.

## 1. Introduction

Homophobic bullying/cyberbullying, in a broad sense, refers to harassment driven by hostility toward the LGBTQI+ community (Lesbian, Gay, Bisexual, Transexual, Queer, Intersex, +). LGBTQI+-phobic bullying/cyberbullying constitutes a subtype of bias-based bullying motivated by actual or perceived sexual orientation and gender identity/expression. Such harassment encompasses face-to-face bullying behaviors (traditional bullying), including physical, verbal, social exclusion, psychological, and sexual aggression (e.g., hitting, insults, marginalization, humiliation, threats, sexual harassment…), as well as cyberbullying or technology-mediated harassment (e.g., spreading false rumors or personal information to damage the victim’s reputation, homophobic or transphobic memes, massive hate attacks, sexual harassment, threats or insults via social media, email, or instant messaging services, dissemination of intimate images or videos without the person’s consent, creation of online groups or communities aimed at isolating and marginalizing the victim).

Victims frequently endure these aggressions perpetrated by one or multiple aggressors. There exists an imbalance of power between the victim and the aggressor(s) whether physical, social, or psychological. The victim is stigmatized, dehumanized, and generally unable to escape this situation on their own. The group of victims includes not only LGBTQI+ youth but also any individual who deviates from dominant sexual norms or binary gender expectations. Aggressors’ violent behaviors are rooted in homophobia, sexism, and values associated with heterosexism. Homophobic bullying/cyberbullying occurs when aggression is based on the victim’s actual or perceived sexual orientation and/or gender identity/expression, positioning that person outside the group’s heteronormativity, regardless of their membership in the LGBTQI+ community. This is a persistent phenomenon that affects the educational system worldwide, and empirical evidence clearly indicates that schools are not perceived as safe environments for sexual and gender minority youth.

### 1.1. Homophobic Bullying/Cyberbullying: Prevalence

Prevalence studies consistently indicate that non-heterosexual individuals, including LGBTQI+ students, and those perceived as non-heterosexual (i.e., individuals who do not conform to traditional masculinity or femininity stereotypes embedded within heteronormative frameworks) experience higher levels of victimization and cybervictimization compared to their heterosexual counterparts [1,2,3,4,5,6,7,8,9,10,11,12,13,14,15,16,17].

Between 50% and 70% of LGBT individuals have experienced bullying related to their sexual orientation or gender identity at some point in their lives [12,14,18,19,20]. A recent UNESCO report concludes that 42% of LGBT (Lesbian, Gay, Bisexual, Transgender) youth worldwide have been mocked, ridiculed, insulted, or threatened at school because of their sexual orientation and/or gender identity or expression [21]. Research on cyberbullying further reveals rates of cybervictimization among LGBT individuals ranging from 10.5% to 71.3% [1,12].

Kosciw et al. (2018) [12] reported notably high prevalence rates of various forms of school bullying among LGBTQ (Lesbian, Gay, Bisexual, Transexual, Queer), students, associated with their sexual orientation, gender expression, and gender. Verbal harassment (e.g., insults, threats) was reported by 70.1% of students due to their sexual orientation, 59.1% due to gender expression, and 53.2% due to gender. Physical harassment (e.g., pushing) was experienced by 28.9%, 24.4%, and 22.8%, respectively. Physical assaults (e.g., punching, kicking, weapon-related injuries) were reported by 12.4%, 11.2%, and 10%, respectively. In addition, 48.7% experienced cyberbullying (via text messages or social media posts), and 57.3% reported sexual harassment (e.g., unwanted touching, sexual comments).

Although bullying and cyberbullying are widespread phenomena in schools, they are more prevalent among LGBTQI+ students. DeSmet et al. (2018) using a sample of students aged 12 to 18, found bullying rates of 27.1% versus 14% and cyberbullying rates of 11.6% versus 7.3% when comparing LGBQ and heterosexual students, respectively [7]. Similarly, Kahle (2020), in another sample within the same age range, reported higher prevalence rates of homophobic bullying (33% versus 6%) among LGBQ youth compared to their heterosexual peers [11]. Garaigordobil and Larrain (2020) [9], in a study with students aged 13 to 17, found victimization rates of 68% versus 37.9% and cybervictimization rates of 53.4% versus 33.9% among non-heterosexual versus heterosexual participants. However, the percentage of heterosexual and non-heterosexual perpetrators and cyberperpetrators was similar.

Angoff and Barnhart (2021), using data from the 2017 Youth Risk Behavior Survey, confirmed a higher risk of victimization among sexual minority youth, with bisexual students being more likely than gay or lesbian students to experience cybervictimization [2]. Similarly, Gámez-Guadix and Incera (2021), in a study with students aged 12 to 18 (8.2% identifying as sexual minorities), found that sexual minority youth frequently experience online sexual violence: 40% reported online sexual violence, 28.4% reported gender-based online violence, 45.2% reported unwanted sexual attention, 9% reported sextortion, and 5.5% reported revenge pornography [22]. Liu et al. (2023), in a study with students aged 15 to 18, found that homosexuality, bisexuality, and uncertainty about sexual orientation were significantly associated with higher levels of bullying and cyberbullying victimization, with homosexual students being at the greatest risk of victimization [23].

Students also perceive the LGBTQ+ community as one of the most vulnerable groups, noting that transgender individuals, in particular, are at greater risk of cybervictimization [24]. Certain gender identities (such as transgender and non-binary individuals) are more frequently targeted in virtual environments, where LGBTQ+-phobic comments are associated with beliefs about gender binarism, stereotypes, gender norms, and broader sociocultural patterns [25]. Homotransphobic discourse proliferates primarily on Twitter (X) and in team sports, particularly football (soccer) and its fan communities, which, together with peer groups, constitute some of the most conducive contexts for the development of such online attitudes [26].

Empirical evidence reveals distinct developmental and gender-specific patterns: an increase in homophobic name-calling [27] and in cybervictimization with age among LGBQ (Lesbian, Gay, Bisexual, Queer) youth, but not among their heterosexual peers [7]. Multiple studies indicate that the school environment constitutes the primary setting where bullying occurs, and that heterosexual cisgender male peers play the most prominent role as perpetrators [7,27,28,29], whereas victims are more frequently girls and non-heterosexual/LGBTQ+ youth [29,30,31].

Discrepancies in prevalence rates across studies can be partially explained by differences in participants’ age ranges, cultural backgrounds, and the specific behaviors assessed, as well as by whether studies measured overall or severe/frequent bullying and cyberbullying. Nevertheless, findings consistently indicate that LGBTQI+ individuals constitute a particularly vulnerable group, experiencing significantly higher levels of victimization and cybervictimization than those with majority, normative sexual orientations and gender identities/expressions.

### 1.2. Homophobic Bullying/Cyberbullying: Connection with Positive Emotional Factors

Studies examining the relationship between homophobic bullying/cyberbullying and positive emotional variables have shown that such victimization negatively impacts self-esteem [19,32,33], which in turn correlates negatively with psychological well-being [34]. LGB (Lesbian, Gay, Bisexual) individuals have been found to report lower levels of personal well-being compared to their heterosexual peers [35,36] and lower life satisfaction [37]. Likewise, LGB identity has been found to correlate negatively with happiness [38], with evidence indicating that this population reports overall lower levels of happiness [39].

On the other hand, individuals with non-normative sexual orientations and gender identities have been found to display higher levels of empathy compared to heterosexual and cisgender populations [40]. Moreover, those scoring higher in empathy were less likely to engage in homophobic verbal harassment [41]. In a one-year longitudinal study, Wright and Wachs (2021) found that adolescents with higher empathy scores were less likely to reproduce homophobic bullying behaviors previously observed among their peers [42]. Similarly, Amadori et al. (2023, 2025) reported that perpetrators of homophobic bullying exhibit low socioemotional competence and suggested that enhancing these competencies among sexual and gender minority youth could help them cope more effectively with bullying situations, underscoring the importance of implementing targeted preventive programs [30,43].

A systematic review of studies examining the relationship between LGBTQI+-phobic bullying/cyberbullying and positive emotional variables confirms that individuals belonging to this population tend to exhibit higher levels of empathy but lower levels of life satisfaction, well-being, happiness, and self-esteem [44]. Nevertheless, it is important to note that only a limited number of studies have explored the associations between homophobic bullying/cyberbullying and emotional intelligence (i.e., emotional attention, clarity, and repair).

### 1.3. Homophobic Bullying: Effects on Development and Mental Health

Victimization and cybervictimization have severe consequences for both developmental processes and mental health. Among the most serious outcomes is the increased risk of suicide; however, these experiences also generate a wide range of academic (e.g., concentration difficulties, decreased school motivation, absenteeism, low academic performance, school failure, and dropout), emotional (e.g., low self-esteem, insecurity, loneliness, unhappiness, guilt, shame, fear, anger, frustration, irritability, and aggressiveness), psychosocial (e.g., introversion, social withdrawal), and mental health problems (e.g., anxiety, depression, post-traumatic stress, and eating disorders among others), many of which may persist into adulthood [45,46].

Although many heterosexual adolescents and young adults report experiences of homophobic victimization and cybervictimization, longitudinal research examining its mental health consequences remains limited. Poteat et al. (2014) followed heterosexual adolescents over the course of an academic year and found that those who experienced homophobic victimization at the beginning of the school year reported higher levels of anxiety and depressive symptoms by the end of the term, particularly among males [47]. The effects were both persistent and specific to homophobic victimization, extending beyond the influence of other forms of bullying. These findings highlight the need to consider the distinct nature of homophobic victimization experienced by young people, including heterosexual youth.

The impact of bullying and cyberbullying on the mental health of LGBTQI+ individuals is devastating. Scientific evidence consistently identifies suicide risk as one of the most severe consequences [1,4,10,48,49,50,51,52,53,54,55]. In parallel, other studies confirm that LGBTQI+ individuals who have experienced bullying and cyberbullying at school report higher levels of depression, anxiety, psychological distress, and emotional strain [19,22,23,28,47,49,52,56,57,58]. Among adolescents, the damage is particularly profound, as it occurs during the process of identity formation, depriving them of safe spaces for self-expression and forcing many to conceal their identities.

Online sexual violence has been associated with poorer mental health outcomes among sexual minorities [22], whereas cyberbullying targeting LGBTQ+ individuals specifically affects emotional well-being, social relationships, and both academic and occupational performance [25]. These effects may persist into adulthood. In a sample of gay and bisexual men aged 20 to 25, Lin et al. (2022) found that all forms of homophobic bullying experienced during childhood were directly associated with borderline personality disorder symptoms in early adulthood, although this association weakened when individuals reported higher levels of family support [59]. Furthermore, LGB youth who express uncertainty about their sexual orientation report higher levels of school bullying, homophobic victimization, substance use, depression, suicidal ideation, and school absenteeism compared to their heterosexual peers or LGB students without such doubts [48]. This heightened vulnerability is also reflected in health-related quality of life, as adolescents questioning their sexual orientation exhibit poorer mood and lower social acceptance when victimized, yet interestingly show better physical well-being when acting as perpetrators [60]. A positive school climate and family support significantly moderate these outcomes [48,59].

Comparative studies confirm greater mental health deterioration among LGBTQI+ victims of bullying/cyberbullying compared to their heterosexual peers, with higher scores on depression (BDI-II) and global psychopathology index (SCL-90). Moreover, victims of bullying exhibited significantly higher levels of social anxiety (SAS) [9]. Similarly, Liu et al. (2023) found an increased risk of emotional problems, particularly among bisexual youth [23]. LGBTQ+ individuals experience significantly more high-risk online interactions compared to heterosexuals, reporting poorer overall mental health and higher rates of self-harm associated with cyberbullying [61]. These patterns of greater mental health deterioration have been observed not only among victims but also among non-heterosexual perpetrators [15].

In conclusion, the evidence consistently shows that victimization and cybervictimization among LGBTQI+ individuals are associated with greater deterioration of mental health compared to that of victims and cybervictims with a majority, normative sexual orientation and gender identity/expression.

Despite the growing body of evidence on the impact of homophobic bullying, important gaps persist in the scientific literature. First, there is a notable geographical concentration of studies in Western Europe and North America, with limited representation of Latin American populations. This geographical limitation is particularly relevant, as attitudes toward sexual diversity and patterns of harassment may vary considerably across cultural contexts. Second, research has primarily focused on victims, paying less attention to perpetrators and bystanders. Finally, very few studies have examined, in an integrated manner, differences in positive emotional variables (emotional intelligence, empathy, happiness) across bullying roles as a function of sexual orientation. These limitations underscore the need for studies addressing these dynamics across diverse cultural contexts using a more comprehensive approach.

### 1.4. Objectives and Hypotheses

This study had four objectives: (1) To identify the prevalence of homophobic bullying (victims, perpetrators, and bystanders) and compare prevalence by sexual orientation and sex; (2) To explore whether there are differences by sexual orientation among victims and perpetrators of homophobic bullying in positive emotional variables (emotional intelligence, empathy, happiness) and psychopathological symptoms; (3) To analyze whether victims and perpetrators of homophobic bullying have sought psychological assistance significantly more often in the past year compared to those who have neither been victims nor perpetrators; and (4) To identify predictive variables of victimization and perpetration in homophobic bullying. Based on these objectives and the literature review, four hypotheses were proposed:

**H1.** 
*A high percentage of students will acknowledge the existence of homophobic bullying behaviors. Approximately 60% will have been victims, 10% perpetrators, and 50% bystanders. A higher percentage of non-heterosexual individuals will be found among victims and bystanders. No significant sex differences will emerge among victims, but a higher proportion of perpetrators will be male.*


**H2.** 
*Non-heterosexual victims, compared to heterosexual victims, will report lower happiness, higher empathy, and higher levels of global psychopathology. Differences between non-heterosexual and heterosexual perpetrators will be smaller; both groups will show low empathy, with non-heterosexual perpetrators exhibiting higher levels of global psychopathology.*


**H3.** 
*Victims and perpetrators of homophobic bullying will have sought psychological assistance significantly more often in the past year for various reasons (e.g., anxiety, depression, eating disorders, violent behavior, and academic problems among others) than those who have neither been victims nor perpetrators.*


**H4.** 
*Identifying as non-heterosexual, exhibiting specific psychopathological symptoms, and reporting lower levels of happiness would predict higher levels of homophobic bullying victimization, whereas being male, showing low empathy, and presenting elevated psychopathology would predict higher levels of homophobic bullying perpetration.*


## 2. Materials and Methods

### 2.1. Participants

The study sample is made up of 1558 adolescents from Cochabamba (Bolivia) aged 13 to 17 years (mean age = 14.64, standard deviation = 0.96) from 18 schools, of whom 50.2% (*n* = 782) were female and 49.8% (*n* = 776) were male. Concerning educational level, 53.7% (*n* = 837) are in 3rd grade of Secondary Education and 46.3% (*n* = 721) are studying 4th grade (54.9% public schools and 45.1% in private schools). The distribution of the sample by sexual orientation was 93.3% heterosexual (*n* = 1453) and 6.7% non-heterosexual (*n* = 105), including 4.2% unsure, 1.9% bisexual, 0.3% lesbian, and 0.3% gay.

The participants constitute a representative sample of the student population in the 3rd and 4th grades of Secondary Education in Cochabamba, Cercado Province, Bolivia. The sample was selected randomly and was representative of the students of the last cycle of Secondary Education of Cochabamba (*n* = 31,895). Using a confidence level of 0.99, with a sample error of 0.03%, the representative sample is 1500. A stratified sampling technique was used to select the sample, taking into account the following parameters: type of school (public–private), educational level (3rd and 4th grades), and sex.

### 2.2. Instruments

To measure the target variables, in addition to a questionnaire designed to collect various sociodemographic data (grade, sex, school type, sexual orientation, request for psychological assistance, etc.), we used 6 standardized instruments with psychometric guarantees of reliability and validity.

*Escala de medición de bullying homofóbico [Homophobic Bullying Measurement Scale]* (EBH) [33]. This scale, designed to assess homophobic bullying, consists of 33 items with a 6-point Likert-type response format (1 = never; 2 = once; 3 = sometimes; 4 = many times; 5 = almost always; 6 = always). The items of the scale assess traditional bullying and cyberbullying behaviors. Each student is required to report the frequency with which they have been involved in each situation related to homophobic bullying. Specifically, the scale allows for the assessment of whether a student has been a victim of homophobic bullying (items 1–26; e.g., “they have laughed at or mocked me because of my sexual orientation”), a perpetrator of homophobic bullying (item 30; e.g., “I have spread a rumor about another classmate’s gender identity/expression or sexual orientation.”), or a bystander of homophobic bullying (items 27–29, 31–33; e.g., “I have supported/accompanied a classmate who has been ignored because of their gender identity/expression or sexual orientation”). Psychometric studies of the scale have shown relatively high item–total correlations within each factor and acceptable internal consistency (α = 0.74). Factor analysis identified three distinct factors. Furthermore, significant correlations were observed between victimization and bystander behavior, as well as between victimization and retrospective self-esteem. Internal consistency for the sample in the present study was high (α = 0.92).

*Trait Meta-Mood Scale* (TMMS-24) [62,63] assesses intrapersonal Emotional Intelligence (EI) using three factors: (1) Attention to Feelings is the amount of attention paid to one’s emotional states. This subscale assesses a basic ability in meta-mood experience referred to the tendency to take notice of and value mood; (2) Emotional Clarity refers to understanding one’s emotional states. This subscale concerns the extent to which people experience their feelings clearly or understand how they feel. It is a relatively enduring tendency to monitor one’s feelings and to experience them lucidly; and (3) Emotional Repair is the ability to regulate one’s emotional states. It refers to the individual’s belief about his/her ability to quit and regulate negative emotional states and to extend positive ones. The sum of all items yields an overall score for intrapersonal emotional intelligence. The test consists of 24 statements (8 for each factor), for example: “I usually worry a lot about what I feel”, “I often become aware of my feelings in different situations,” and “When I am angry, I try to change my mood.” The participants must respond using a 5-point Likert scale ranging from 1 (strongly disagree) to 5 (strongly agree). The reliability of the scale (Cronbach’s alpha) is high (attention α = 0.90; clarity α = 0.90; repair α = 0.86). The test–retest correlations between the two applications after four weeks were satisfactory: attention (r = 0.60), clarity (r = 0.70), and repair (r = 0.83). The dimensions of emotional intelligence show significant associations with each other. Validity studies have shown positive correlations with life satisfaction, and negative correlations with depression and rumination. The internal consistency obtained with the sample of this study was very high for the total scale (α = 0.95) and for the subscales (attention α = 0.91, clarity and repair α = 0.92).

*Test de Empatía Cognitiva y Afectiva* [*Cognitive and Affective Empathy Test*] (TECA) [64]. It measures empathy across 4 dimensions: (1) Perspective Taking (the intellectual or imaginative ability to put oneself in another’s place; for example, “before making a decision I try to take all perspectives into account”); (2) Emotional Comprehension (the ability to recognize and understand other people’s moods, intentions, and impressions; for example, I can easily tell when someone is in a bad mood); (3) Empathic Distress (the ability to share another person’s negative emotions; for example, I feel sad just because a friend is sad); and (4) Empathic Joy (the ability to share another person’s positive emotions; for example, I feel good when others are having fun). The sum of all the items yields an overall score for cognitive and affective empathy. The TECA consists of 33 items on which participants rate their degree of agreement on a Likert response format ranging from 1 (strongly disagree) to 5 (strongly agree). Validity was demonstrated through its significant relationships with other instruments assessing empathy, such as the Interpersonal Reactivity Index (IRI). The internal consistency obtained with the original sample was adequate (α = 0.86), and with the sample of the present study it was high (α = 0.91).

*The Oxford Happiness Questionnaire* [65,66]. The OHQ was derived from the Oxford Happiness Inventory (OHI), which reduced to 29 items attempts to measure the happiness of a general nature of each individual, that is, psychological well-being. For example, “I am not particularly optimistic about the future,” “I am well satisfied about everything in my life,” “I am very happy,” “Life is good,” and “I always have a cheerful effect on others”. The person expresses his or her degree of agreement with the statements on a 6-point Likert scale (1 = strongly disagree; 6 = strongly agree). In the original study, the associations of the OHI and the OHQ were compared, obtaining significant correlations that support construct validity. The studies carried out with a sample of people aged between 13 and 68 verified the good reliability of this scale (α = 0.91) based on standardized items. The Spanish adaptation with adolescents showed good internal consistency (α = 0.86), the same as that obtained with the sample of the present study (α = 0.97).

*Social Anxiety Scale for Adolescents* (SAS) [67,68]. Composed of 22 items, the instrument assesses overall social anxiety (social phobia) and three subdimensions: (1) Fear of negative evaluation (e.g., “I worry about being judged by others”); (2) Social avoidance and distress in situations involving unfamiliar people (e.g., “I feel nervous when I am introduced to strangers”); and (3) Social avoidance and distress in the company of acquaintances (e.g., “I feel embarrassed even when I am with people I know well”). The sum of the three subscale scores yields a composite score on General Social Avoidance and Distress. Adolescents report the frequency with which they experience these thoughts, feelings, and behaviors on a 5-point Likert scale (1 = never, 2 = rarely, 3 = sometimes, 4 = often, 5 = always). The SAS has been psychometrically validated in Spanish adolescent populations, supporting the original three-factor structure. The internal consistency of the scale in the Spanish adaptation sample was high (α = 0.91), as was that obtained in the present study (α = 0.92).

*Symptoms Checklist-90-Revised* (SCL-90-R) [69,70]. It contains 90 items distributed on 9 scales that report on psychopathological disorders: somatization (experiences of body dysfunction, neurovegetative alterations of the cardiovascular, respiratory, gastrointestinal and muscular systems), obsession-compulsion (absurd and unwanted behaviors, thoughts, etc., that generate intense distress and are difficult to resist, avoid, or eliminate), interpersonal sensitivity (timidity and embarrassment, discomfort and inhibition in interpersonal relationships), depression (anhedonia, hopelessness, helplessness, lack of energy, self-destructive ideas, etc.), anxiety (generalized and acute anxiety/panic), hostility (aggressive thoughts, feelings and behaviors, anger, irritability, rage, and resentment), phobic anxiety (agoraphobia and social phobia), paranoid ideation (paranoid behavior, suspicion, delirious ideation, hostility, grandiosity, need for control, etc.), and psychoticism (feelings of social alienation). Furthermore, the test makes it possible to calculate the Global Severity Index (GSI), which is a standard and indiscriminate measure of the intensity of global psychopathological suffering. Adolescents report the frequency with which they have experienced these symptoms during the last month using a Likert scale ranging from 0 to 4 (0 = never, 1 = somewhat, a little, 2 = moderately, 3 = quite a lot, 4 = very much). Sample items include: “headaches,” “having to check everything they do repeatedly,” “feeling inferior to others,” “suicidal thoughts or ideas of ending one’s life,” “sudden terror or panic attacks,” “outbursts of anger or uncontrollable fits of rage,” “fear of open spaces or being outdoors,” “the impression that most of their problems are caused by others,” and “hearing voices that other people do not hear.” Studies with Spanish samples suggest good reliability (α = 0.90), as in this study (α = 0.95). Specifically, the reliability of each subscale was the following: somatization (α = 0.92); obsession-compulsion (α = 0.92); interpersonal sensitivity (α = 0.91); depression (α = 0.92); anxiety (α = 0.94) hostility (α = 0.90); phobic anxiety (α = 0.90); paranoid ideation (α = 0.89) and psychoticism (α = 0.92).

### 2.3. Procedure

This study uses a descriptive and comparative cross-sectional methodology. Firstly, a letter was sent to the headmasters of the randomly selected schools, explaining the research project. Those who agreed to participate received informed consent for parents and participants. When the director of the selected center refused to collaborate, the procedure was repeated with the next center on the list, taking into account the type (public–private) of the center that declined to participate. Subsequently, the evaluation team visited the schools and administered the assessment tools to the students (in two 40-min sessions).

The study met the ethical values required in research with human beings, respecting the fundamental principles included in the Helsinki Declaration, in its latest version, and in the active rules: informed consent and right to information, protection of personal data, and guarantees of confidentiality, non-discrimination, gratuity, and the possibility of dropping out of the study in any of its phases. This study received the favorable report of the Ethics Committee of the University of the Basque Country (CEISH-UPV/EHU:M10_2017_094MR1_Garaigordobil Landazabal).

### 2.4. Data Analysis

First, in order to identify the prevalence of homophobic bullying, the percentages of students who reported having experienced, perpetrated, or witnessed any of the 33 behaviors assessed by the Homophobic Bullying Scale (EBH) were analyzed (overall prevalence = once or more; severe prevalence = many times + almost always + always). Additionally, the percentages of victims, perpetrators, and bystanders of homophobic bullying were calculated at a global level (once or more), and comparisons were carried out according to sexual orientation (heterosexual vs. non-heterosexual) and sex (male vs. female). Pearson’s chi-square test was applied to examine group differences.

Second, to explore whether differences exist among victims and perpetrators of homophobic bullying according to sexual orientation in relation to positive emotional variables (emotional intelligence, empathy, and happiness) and psychopathological symptoms, multivariate analyses of variance (MANOVA) were conducted. Moreover, for each variable, descriptive (means and standard deviations) and univariate analyses (ANOVA) were performed, and effect sizes were calculated (Cohen’s d: small < 0.50; moderate 0.50–0.79; large ≥ 0.80).

Subsequently, to determine whether victims and perpetrators had sought psychological assistance in the past year significantly more often than those who had neither been victims nor perpetrators of homophobic bullying, contingency analyses were carried out, computing frequencies, percentages, and Pearson’s chi-square for both roles.

Finally, to identify the predictor variables of homophobic bullying (victimization/perpetration), hierarchical regression analyses were conducted. Variables were entered in theoretically driven blocks following this order: Block 1 (Model 1): demographic variables (age, sex, sexual orientation); Block 2 (Model 2): positive emotional variables (emotional intelligence, empathy, happiness); and Block 3 (Model 3): psychopathological variables (social anxiety and psychopathological symptoms). Demographic variables were entered first to control for their effects, as previous studies have documented differences in bullying based on sexual orientation [9] and gender [27]. Emotional variables were included in the second block, drawing on evidence suggesting that factors such as empathy, emotional intelligence, and happiness function as protective factors against victimization and perpetration [43,44]. Finally, psychopathological symptoms were added in the third block to examine whether mental health difficulties predicted victimization and perpetration beyond the effects of demographic and emotional variables [23,56]. Separate hierarchical regression models were conducted for victimization and perpetration as dependent variables.

## 3. Results

### 3.1. Homophobic Bullying: Prevalence and Comparison by Sexual Orientation and Gender

The prevalence results for homophobic bullying across the 33 behaviors assessed with the EBH Homophobic Bullying Scale (see Table 1) indicate that a substantial proportion of students reported having experienced one or more instances of homophobic bullying. Five behaviors emerged as the most prevalent: “My classmates have criticized my expressions, ways of speaking, or behaving” (26.8%); “I have been criticized for my aesthetic choices (clothing, hairstyle, makeup, etc.)” (29.8%); “They have laughed at or mocked me because of my gender identity or expression” (31%); “My classmates have done things to bother me (throwing things, blocking my way, pushed me, etc.)” (36.2%); and “I have been given derogatory, degrading, or offensive nicknames because of my gender identity or expression” (39%). In contrast, 11.8% of participants reported having engaged in behaviors reflecting perpetration of homophobic bullying, such as “I have spread a rumor about another classmate’s gender identity/expression or sexual orientation.” Regarding bystanding behaviors, although 20.6% of students acknowledged having taken a passive stance (e.g., “If someone was hit because of their gender identity/expression or sexual orientation, I did nothing”), a larger proportion reported having taken a more positively active role as bystanders. Specifically, 30.9% stated that “If someone bothered a classmate because of their gender identity/expression or sexual orientation, I intervened to stop it”; 35.2% reported “If someone laughed at a classmate because of their gender identity/expression or sexual orientation, I tried to stop it”; and 37.7% affirmed “I have supported/accompanied a classmate who has been ignored because of their gender identity/expression or sexual orientation.”

Complementarily, the overall prevalence results confirm that a considerable proportion of students reported experiences of victimization, perpetration, and bystanding in homophobic bullying (see Table 2).

Victims: 76.6% reported having experienced one or more homophobic aggressive behaviors by peers at some point in their lives. The percentage of victims by sexual orientation was 75.8% heterosexuals and 87.5% non-heterosexuals, with statistically significant differences by sexual orientation (χ^2^ = 7.40, *p* < 0.01). The percentage of victims by gender was 76.5% males and 76.7% females, with no significant gender differences (χ^2^ = 0.013, *p* > 0.05). Therefore, non-heterosexual students experienced significantly more homophobic bullying, while victimization rates were similar for both genders.

Perpetrators: 11.8% reported having perpetrated one or more homophobic aggressive behaviors toward peers at some point in their lives. The percentage of perpetrators by sexual orientation was 11.7% heterosexuals and 13.5% non-heterosexuals, with no statistically significant differences (χ^2^ = 0.306, *p* > 0.05). However, when considering gender, 14.2% of males and 9.4% of females reported perpetrating homophobic bullying, with statistically significant gender differences (χ^2^ = 8.49, *p* < 0.01). Thus, the percentage of heterosexual and non-heterosexual perpetrators was similar, while the percentage of male perpetrators was significantly higher than the percentage of female perpetrators.

Bystanders: 51.9% reported having witnessed one or more homophobic aggressive behaviors inflicted by peers at some point in their lives. The percentage of bystanders by sexual orientation was 51% heterosexuals and 64.4% non-heterosexuals, showing statistically significant differences (χ^2^ = 7.03, *p* < 0.01). By gender, 49% of males and 54.7% of females reported having witnessed such behaviors, also with statistically significant differences (χ^2^ = 4.98, *p* < 0.05). Therefore, non-heterosexual students and females were significantly more likely to witness homophobic bullying than heterosexual students and males, respectively.

### 3.2. Emotional Factors and Psychopathological Symptoms in Victims and Perpetrators of Homophobic Bullying: Differences by Sexual Orientation

Box’s M and multivariate normality tests were not provided by SPSS (version 24) due to singular covariance matrices in the smaller group and the large number of dependent variables. Given the robustness of MANOVA to these issues, Pillai’s Trace was used as the primary statistic. To examine whether differences existed between the two profiles (heterosexual vs. non-heterosexual) among victims across the study variables, a MANOVA was conducted using the overall set of scores. Results revealed significant multivariate differences by profile, Pillai’s Trace = 0.057, F(20, 1163) = 3.44, *p* < 0.001, and the effect size was small (η^2^ = 0.057, *r* = 0.238). Descriptive and variance analyses between the two profiles (see Table 3) showed that non-heterosexual victims (compared to heterosexual victims) reported significantly lower scores in emotional regulation, empathic joy, overall empathy, and happiness, as well as higher scores in fear of negative social evaluation, general social avoidance and distress, and in all psychopathological symptoms assessed (somatization, obsession–compulsion, interpersonal sensitivity, depression, anxiety, hostility, phobic anxiety, paranoid ideation, and psychoticism), together with a higher global psychopathology index (GSI). Effect sizes were medium for the psychopathological symptoms and small for the remaining variables.

Beyond statistical significance, effect sizes revealed clinically meaningful differences between heterosexual and non-heterosexual victims across most psychopathological dimensions. For example, depression showed a moderate effect size (d = 0.59), meaning that non-heterosexual victims scored approximately 0.6 standard deviations higher than heterosexual victims. Similar effect sizes were observed for somatization (d = 0.58), obsession–compulsion (d = 0.56), psychoticism (d = 0.56), anxiety (d = 0.50), and the Global Severity Index (GSI) (d = 0.60), all falling within the moderate range. These values highlight not only statistical significance but also substantial clinical relevance in the heightened psychopathology reported by non-heterosexual victims. In contrast, effect sizes for emotional variables were small (d = 0.20–0.30), reflecting only modest differences between groups, consistent with typical findings in constructs related to emotional well-being. Overall, the pattern of effect sizes underscores that the psychological burden experienced by non-heterosexual victims is not only statistically detectable but also meaningful in terms of magnitude.

To examine whether differences existed between the two profiles (heterosexual vs. non-heterosexual) among perpetrators, a second MANOVA was performed using the same variables. The results indicated significant multivariate differences by profile, Pillai’s Trace = 0.191, F(20, 161) = 1.84, *p* < 0.05, and the effect size was medium (η^2^ = 0.191, *r* = 0.437). Descriptive and variance analyses between the two profiles (see Table 3) showed that non-heterosexual perpetrators (compared to heterosexual perpetrators) presented significantly higher scores only in some psychopathological symptoms (depression, anxiety, hostility, paranoid ideation, and psychoticism) and in the global psychopathology index (GSI), with medium effect sizes. Therefore, the results indicate few differences between heterosexual and non-heterosexual perpetrators overall.

For perpetrators, the pattern of effect sizes differed. Although fewer significant group differences emerged between heterosexual and non-heterosexual perpetrators, those that did appear tended to show moderate effect sizes. The largest differences were found in depression (d = 0.58), psychoticism (d = 0.57), hostility (d = 0.55), paranoid ideation (d = 0.55), and anxiety (d = 0.51) reflecting differences of 0.5 to 0.6 standard deviations. These results point to medium-sized effects that are relevant for the clinical interpretation of the psychological profile of this subgroup of perpetrators. In contrast, effect sizes for emotional competence and well-being variables were small (d < 0.30), suggesting broadly similar emotional profiles among heterosexual and non-heterosexual perpetrators.

### 3.3. Psychological Help-Seeking Among Victims and Perpetrators of Homophobic Bullying

Contingency analyses (see Table 4) showed that victims of homophobic bullying were significantly more likely to have sought psychological assistance in the past year than non-victims (χ^2^ = 8.19, *p* = 0.017). Among those who had sought psychological help for various problems (e.g., anxiety, depression, academic difficulties, eating problems, violence-related issues), 80.8% were victims and 19.2% were non-victims. Logistic regression analyses confirmed that victims had significantly higher odds of seeking psychological help (OR = 1.392, 95% CI [1.04, 1.86], *p* = 0.024), representing a 39% increase in the likelihood of help-seeking. Similarly, perpetrators of homophobic bullying showed higher rates of psychological help-seeking (15.2%) compared to non-perpetrators (10.8%) (χ^2^ = 6.60, *p* = 0.037). Logistic regression analyses revealed that perpetrators had significantly higher odds of seeking psychological help (OR = 1.507, 95% CI [1.07, 2.10], *p* = 0.017), representing a 51% increase compared to non-perpetrators. In summary, both victims and perpetrators of homophobic bullying reported significantly higher rates of psychological help-seeking in the past year compared to their counterparts who had not been involved in such behaviors.

### 3.4. Predictive Variables of Victimization and Perpetration of Homophobic Bullying

A hierarchical multiple regression was conducted to examine predictors of Homophobic Bullying Victimization (see Table 5). In Step 1, demographic covariates (sex, age, and sexual orientation) explained a small but significant portion of variance in victimization, R^2^ = 0.024, F(3, 1503) = 12.40, *p* < 0.001. In Step 2, the addition of emotional competence and well-being measures produced a further significant, albeit modest, increase in explained variance, ΔR^2^ = 0.028, ΔF(8, 1495) = 5.51, *p* < 0.001, resulting in R^2^ = 0.052. Finally, Step 3 added psychopathology and social-anxiety scales, which yielded the largest increment, ΔR^2^ = 0.134, ΔF(12, 1483) = 20.31, *p* < 0.001; the full model explained R^2^ = 0.186 of the variance in victimization (Adjusted R^2^ = 0.173), F(23, 1483) = 14.721, *p* < 0.001. In the final model, being non-heterosexual was associated with higher victimization (B = 4.05, SE = 1.12, β = 0.087, *p* < 0.001), as was being male (B = −1.83, SE = 0.61, β = −0.077, *p* = 0.003). Among psychological variables, somatization (B = 1.44, SE = 0.51, β = 0.106, *p* = 0.005), paranoid ideation (B = 1.58, SE = 0.59, β = 0.119, *p* = 0.008) and fear of negative evaluation (B = 0.15, SE = 0.06, β = 0.103, *p* = 0.008) were positive predictors, while overall happiness was negatively associated with victimization (B = −0.027, SE = 0.01, β = −0.080, *p* = 0.008). Other predictors were non-significant once all variables were included.

Additionally, a hierarchical multiple regression was conducted to examine predictors of Homophobic Bullying Perpetration (see Table 6). Demographic covariates (sex, age, and sexual orientation) in Step 1 explained a small but significant proportion of the variance (R^2^ = 0.007, F(3, 1503) = 3.56, *p* = 0.014), with sex emerging as the only significant predictor in this block (B = −0.069, SE = 0.034, β = −0.053, *p* = 0.041). In Step 2, emotional competences and well-being variables produced an additional small increase in explained variance (ΔR^2^ = 0.015, ΔF(8, 1495) = 2.833, *p* = 0.004), yielding R^2^ = 0.022. Only emotional attention was significant (B = 0.008, SE = 0.003, β = 0.113, *p* = 0.001). Finally, adding psychopathology and social anxiety measures in Step 3 significantly improved the model (ΔR^2^ = 0.029, ΔF(12, 1483) = 3.80, *p* < 0.001), with the full model explaining 5.1% of the variance in perpetration (Adjusted R^2^ = 0.036), F(23, 1483) = 3.467, *p* < 0.001. In the final model, sex emerged as a significant predictor (B = −0.105, SE = 0.036, β = −0.081, *p* = 0.004), indicating that males showed higher perpetration scores. Phobic anxiety was also a significant positive predictor (B = 0.063, SE = 0.030, β = 0.090, *p* = 0.034). No other psychological variable reached statistical significance when entered simultaneously.

## 4. Discussion

First, the findings reveal that a substantial proportion of students acknowledge the existence of multiple forms of homophobic bullying (insults, humiliation, aggression, mockery, and homophobic name-calling). Specifically, 76.6% reported having experienced homophobic behaviors (with a higher percentage of non-heterosexual victims and similar rates across sexes), 11.8% admitted having engaged in homophobic aggressive behaviors toward peers (with a higher proportion of male perpetrators and no significant differences by sexual orientation), and 51.9% reported having witnessed homophobic acts (with a higher percentage of non-heterosexual and female observers). These findings confirm H1 and are consistent with previous studies documenting the high prevalence of homophobic bullying and the increased vulnerability of non-heterosexual individuals [8,9,12].

The present results: (1) demonstrate a high prevalence of homophobic bullying affecting both non-heterosexual and heterosexual students, confirming previous findings [8,14]; (2) corroborate that the percentage of victims and cybervictims of homophobic bullying is considerable, as shown in other studies [9,11,12,18]; and (3) support prior research indicating that non-heterosexual individuals experience more bullying and cyberbullying behaviors compared to their heterosexual peers [1,2,3,4,5,6,7,9,11,12,13,15,16,17], with heterosexual males being the most frequent perpetrators [28,29,31].

Contrary to several previous studies reporting higher rates of victimization among girls and non-heterosexual/LGBTQ+ youth [29], our findings did not reveal significant gender differences among victims of homophobic bullying. This discrepancy may reflect specific characteristics of our sample, including regional or cultural variations, age range, or methodological differences in measurement. However, consistent with prior research [7,27,28,29], we observed a significantly higher proportion of male perpetrators, supporting the notion that males, particularly heterosexual cisgender males, are more frequently involved in perpetration. Future research should further investigate contextual factors that might explain variations in gender patterns of victimization across different cultural settings.

Although this study addresses the identified need for research in Latin American contexts, it is important to recognize that the findings stem from a specific cultural context. Bolivia presents a paradox between a progressive legal framework, including constitutional protection against discrimination based on sexual orientation and gender identity since 2009 and Law 807 on gender identity, and markedly conservative social attitudes [71]. The country exhibits one of the lowest levels of LGBTI acceptance in Latin America (acceptance index 5.24, ranking 55th globally), with acceptance levels stagnating since 2000 [72]. Bolivia also shows among the lowest levels of contact with LGB individuals and some of the most negative attitudes toward sexual diversity in Latin America [73]. Moreover, constructions of hegemonic masculinity strongly associated with homophobia prevail: approximately one-third of Bolivians (35.8% of men, 35.3% of women) endorse the statement that “a homosexual is not a real man,” and between 9% and 16% justify violence toward homosexual men in various situations [74]. This context is further compounded by the limited implementation of comprehensive sexuality education, facing social resistance and resource scarcity [75,76,77], and by high levels of gender-based violence reflecting a broader patriarchal culture [76].

The higher prevalence observed in this study (76.6%) compared to UNESCO’s global 42% [21] may reflect this combination of highly homophobic attitudes, rigid constructions of masculinity, and the absence of effective comprehensive sexuality education. These findings underscore the importance of conducting research in contexts with moderate-to-low levels of LGBTI acceptance, where the rates and dynamics of homophobic bullying may differ substantially from those reported in studies concentrated in Western Europe and North America.

Second, the findings indicate that non-heterosexual victims scored significantly lower in emotional repair, empathic joy, overall empathy, and happiness, while scoring higher in fear of negative social evaluation, general social avoidance and distress, all psychopathological symptoms assessed, and in the global psychopathology index (GSI). Fewer differences were observed between heterosexual and non-heterosexual perpetrators, although non-heterosexual perpetrators showed significantly higher levels of certain psychopathological symptoms (depression, anxiety, hostility, paranoid ideation, psychoticism) and in the global psychopathology index (GSI). These results partially confirm H2. As predicted, non-heterosexual victims displayed lower levels of happiness [38,39] and higher levels of psychopathology, consistent with prior research showing that non-heterosexual individuals experience more depression and anxiety [19,23,47,49,52,56,57,58], psychological distress, and stress [78], as well as higher levels of psychopathological symptoms and overall psychopathology [9,15,22]. However, higher empathy was not observed, contrasting with [40]. This discrepancy may stem from methodological differences: Kleiman et al. (2015) analyzed only male participants, focused on racial (rather than general) empathy, and did not link it to victimization [40]. It is plausible that experiences of bullying limit the social interactions necessary for empathy development. These findings align with Amadori et al. (2025), who emphasize the need to promote socioemotional skills among sexual minorities to foster coping with bullying situations [30]. Furthermore, the results confirm that perpetrators tend to show lower empathy, consistent with Poteat and Espelage (2005), who found that individuals with higher empathy engaged in fewer homophobic insults [41].

Third, the results indicate that both victims and perpetrators of homophobic bullying were significantly more likely to have sought psychological assistance in the past year for various problems (anxiety, depression, eating disorders, violent behavior, school difficulties, etc.) than those who had not been involved in bullying. Thus, H3 was fully confirmed and supports previous findings showing that LGBTQI+ individuals report more mental health and school-related problems [25,48], leading to greater use of psychological services.

Finally, hierarchical regression analyses yielded distinct patterns of significant predictors for homophobic bullying victimization and perpetration. In the victimization model, six variables emerged as significant in the final model: identifying as non-heterosexual, being male, higher levels of somatization, paranoid ideation, and fear of negative evaluation, as well as lower levels of happiness. The full model explained 18.6% of the variance. In contrast, the perpetration model identified only two significant predictors: being male and phobic anxiety. The full model explained 5.1% of the variance, suggesting that perpetration is influenced by a narrower set of factors than victimization.

These results provide partial support for H4. As hypothesized, identifying as non-heterosexual, exhibiting specific psychopathological symptoms (somatization, paranoid ideation, fear of negative evaluation), and reporting lower happiness significantly predicted victimization. However, being male also emerged as a significant predictor of victimization, a finding not initially hypothesized but consistent with constructions of hegemonic masculinity prevalent in the Bolivian context [74,75,76,77].

For perpetration, being male was confirmed as a significant predictor, consistent with H4 and prior research [7,27,28,29]. However, contrary to our hypothesis, empathy did not emerge as a significant predictor when other variables were controlled. Only phobic anxiety (a specific dimension of psychopathology) predicted perpetration, rather than the broader pattern of elevated psychopathology initially hypothesized. This suggests that the relationship between mental health and perpetration may be more nuanced than previously assumed, with only certain anxiety-related symptoms playing a role.

This study makes several significant contributions: (1) It provides evidence of the high prevalence of homophobic bullying targeting both non-heterosexual and heterosexual adolescents in Bolivia. Although many studies have concluded that LGBTQI+ individuals experience higher rates of bullying and cyberbullying, few have specifically analyzed homophobic bullying/cyberbullying as conducted in this study. (2) It offers novel insights into the associations between homophobic bullying and emotional intelligence, empathy, and happiness, comparing homophobic bullying victims and perpetrators on these emotional variables in both heterosexual and non-heterosexual samples. (3) It confirms the findings of previous research showing that non-heterosexual victims, compared to their heterosexual peers, display higher levels of psychopathological symptoms and mental health problems resulting from bullying/cyberbullying victimization, as evidenced by psychometric data (SCL-90, SAS) and self-reports of psychological assistance requests. (4) It identifies relevant predictive variables that highlight the importance of developing socio-emotional competencies and promoting mental health among both victims and perpetrators.

However, the study also has limitations. First, its cross-sectional design prevents causal inferences; longitudinal studies are needed to examine temporal relationships. Second, self-report measures may introduce social desirability bias; future research should triangulate findings with peer and teacher reports. Third, perpetration of homophobic bullying was assessed using a single item, limiting measurement comprehensiveness; findings related to perpetrators should be interpreted cautiously. Fourth, although the instruments have been validated in Spanish-speaking populations, confirmatory validation studies and composite reliability measures (e.g., Omega) specific to Bolivian adolescent samples are lacking and should be addressed in future research.

The results have important practical implications. LGBTQI+-phobic attitudes remain a pressing social issue that requires urgent attention through a multidisciplinary approach encompassing education, prevention, and multidirectional intervention (society, school, family, and clinical context).

The school context is the setting in which most bullying/cyberbullying behaviors against LGBTQI+ students occur. The messages conveyed in educational institutions (through audiovisual materials, teaching resources, and discourse) continue to reflect a heteronormative framework, overlooking the essential need for education grounded in sexual diversity. It is necessary to ensure inclusive curricula free from stereotypes and to promote respect and non-discrimination at all educational levels. Likewise, comprehensive training for teachers and school staff on inclusive practices is essential to foster a climate in which LGBTQI+ students feel safe and comfortable expressing themselves freely.

The findings suggest implementing programs to foster socio-emotional competence development, that is, to integrate Social and Emotional Learning (SEL) into educational curricula [30,43,79], through activities promoting non-violent communication, constructive conflict resolution, respect for differences, empathy, and emotional regulation. These skills increase students’ capacity to prevent and cope with homophobic behaviors and inhibit such behaviors among perpetrators. In parallel, systematic implementation of anti-bullying programs addressing bullying and cyberbullying (including activities focused on harassment due to non-majority or non-normative sexual orientation, identity, or gender expression) is necessary [80,81,82]. Future research should develop anti-bullying programs aimed at challenging and redefining gender and heteronormative beliefs that sustain homophobic bullying and cyberbullying, particularly among heterosexual adolescent boys. Such programs should promote critical reflection on masculinity and sexuality within educational contexts and peer relationships [30].

Families must be actively involved in preventing violence, ensuring that efforts extend beyond the school environment. Collaborative action among all educational agents will contribute to creating schools and learning environments where everyone feels safe, respected, and empowered to learn and thrive. Young people who perceive greater family support are less likely to be involved in bullying behaviors, either as victims or perpetrators [83]. The family context plays a crucial role in the development of ethical-moral values, prosocial behavior, empathy, and respect for diversity. Parents who model prosocial, empathic, and diversity-respectful behaviors (and reinforce these in their children) tend to raise more tolerant, less violent, and more respectful children. Socio-emotional and digital education, which should begin within the family, is essential. Therefore, training programs for families to promote these values and behaviors in their children are necessary [84]. Nevertheless, once bullying/cyberbullying has occurred, another crucial context for intervention is the clinical-therapeutic setting. In light of the findings, it is important to develop accessible clinical interventions aimed at reducing depression, stress, anxiety, and suicide risk among victims of homophobic bullying and cyberbullying.

Intervention at the societal level is also fundamental. For example, media campaigns can raise awareness about the severe consequences of bullying, the importance of respect for differences, and tolerance toward diversity, thereby contributing to changing prevailing social norms. The stereotypes and prejudices sustained by a heteronormative society stigmatize LGBTQI+ individuals and legitimize or promote their victimization. As a society, we have the responsibility to build inclusive and safe physical and digital environments where all people can express their identities without fear of being harassed or discriminated against. This is not only a matter of rights, it is a matter of humanity and respect. To this end, technology companies can also contribute by making digital spaces safer, for instance, by developing machine learning models to identify bullying/cyberbullying content (particularly that targeting LGBTQI+ individuals) to assist social media platforms and online communities in moderating and addressing harmful content directed at vulnerable groups [84]. Collaboration among schools, families, and LGBTQI+ advocacy organizations should also be encouraged to create a comprehensive support network for these individuals.

At the legislative level, many countries still lack specific laws protecting individuals from bullying/cyberbullying in general, and even fewer have laws focused on protecting LGBTQI+ persons. It is crucial to highlight this legal gap and advance the legal regulation of bullying/cyberbullying and of discriminatory or hate-based behaviors targeting LGBTQI+ individuals, ensuring these are classified as criminal offenses. Governments should prohibit and address discrimination by repealing discriminatory laws, banning discrimination against LGBTQI+ individuals, and legally recognizing the gender identity of transgender persons in official documents.

## 5. Conclusions

This study provides evidence of the high prevalence of homophobic bullying among Bolivian adolescents, affecting both heterosexual and non-heterosexual students, as three out of four students reported having experienced such behaviors, with significantly greater vulnerability observed among non-heterosexual youth. The findings demonstrate that non-heterosexual victims exhibit a considerably more compromised mental health profile, characterized by higher psychopathological symptoms, lower emotional regulation, reduced happiness, and greater social anxiety compared to their heterosexual peers.

The identification of variables associated with victimization and perpetration, particularly the global psychopathology index, empathy, happiness, and fear of negative social evaluation, contributes knowledge regarding factors related to homophobic bullying and guides the development of more specific and effective prevention and intervention strategies.

The results underscore the urgent need to implement comprehensive psychoeducational programs in educational settings that combine the development of socio-emotional competencies with specific anti-bullying strategies, focusing on promoting tolerance toward sexual diversity. The practical implications extend beyond the school context, requiring a multidirectional response involving schools, families, mental health professionals, policymakers, and society at large to create safe environments where all adolescents, regardless of their sexual orientation or gender identity, can develop without fear of discrimination. Future research should employ longitudinal designs to deepen understanding of these relationships and develop evidence-based interventions specifically designed to reduce homophobic bullying and mitigate its mental health consequences.

## Figures and Tables

**Table 1 healthcare-13-03119-t001:** Percentage of responses on the 33 items of the Homophobic Bullying Scale.

Items	Never	Once	Sometimes	Many Times	Almost Always	Always	Global	Severe
They have laughed at or mocked me because of my gender identity or expression.	69.0	17.2	9.8	2.1	0.7	1.2	31.0	4
2. They have laughed at or mocked me because of my sexual orientation.	90.6	5.2	2.7	1.0	0.3	0.3	9.4	1.6
3. I have been insulted because of my gender identity or expression.	84.3	8.5	4.9	1.2	0.5	0.5	15.7	2.2
4. I have been insulted because of my sexual orientation.	91.8	4.5	2.5	0.6	0.5	0.1	8.2	1.2
5. My classmates have embarrassed me in front of others because of my gender identity or expression.	83.1	10.1	3.8	1.8	0.9	0.4	16.9	3.1
6. My classmates have embarrassed me in front of others because of my sexual orientation.	90.9	5.3	2.5	0.9	0.3	0.2	9.1	1.4
7. My classmates have done things to bother me (thrown things at me, blocked my way, pushed me, etc.).	63.8	18.4	11.8	3.3	1.2	1.5	36.2	6
8. My classmates have hit me because of my gender identity or expression.	93.3	3.5	2.2	0.6	0.3	0.1	6.7	1
9. My classmates have hit me because of my sexual orientation.	94.9	2.3	1.4	0.7	0.4	0.3	5.1	1.4
10. My classmates have criticized my expressions, ways of speaking, or behaving.	73.2	13.8	7.4	2.5	1.4	1.7	26.8	5.6
11. I have been given derogatory, degrading, or offensive nicknames because of my gender identity or expression.	61.0	22.5	8.9	3.6	1.3	2.7	39.0	7.6
12. I have been given derogatory, degrading, or offensive nicknames because of my sexual orientation.	87.1	6.8	3.6	1.5	0.5	0.6	12.9	2.6
13. I have suffered injuries or serious harm because other classmates have assaulted me.	83.1	10.0	5.0	0.9	0.4	0.6	16.9	1.9
14. I have been ignored because of my sexual orientation.	93.3	3.1	2.3	0.9	0.3	0.2	6.7	1.4
15. I have been ignored because of my gender identity or expression.	90.3	5.2	2.7	1.2	0.5	0.2	9.7	1.9
16. I have been criticized for my aesthetic choices (clothing, hairstyle, makeup, etc.).	70.2	15.7	8.3	2.3	2.3	1.2	29.8	5.8
17. I have been mocked for relating better to people of the opposite gender.	76.3	10.4	6.9	2.8	2.1	1.6	23.7	6.5
18. False rumors have been spread about my gender identity/expression or sexual orientation.	79.2	11.2	5.6	1.8	1.0	1.2	20.8	4
19. I have been criticized for not participating in activities typically associated with my gender (sports, arts, academic, etc.).	76.5	13	6.5	2.2	0.8	1.1	23.5	4.1
20. I have been criticized for excelling in activities not typically associated with my gender (sports, arts, academic, etc.).	81.0	10.0	5.3	2.1	0.7	0.8	19	3.6
21. I have been intimidated with sexual comments or insults.	87.7	6.9	3.2	1.2	0.6	0.5	12.3	2.3
22. I have skipped classes to avoid being harassed.	88.7	6.0	3.4	1.3	0.2	0.4	11.3	1.9
23. I have skipped extracurricular activities (sports, arts, etc.) to avoid being harassed.	89.0	5.9	3.2	1.1	0.4	0.5	11.0	2
24. I have made excuses to miss class out of fear of being harassed.	89.0	6.3	2.5	1.0	0.6	0.6	11.0	2.2
25. I have received threats through devices (internet, phone, mobile, etc.) because of my gender identity or expression.	90.7	5.1	2.7	0.7	0.5	0.3	9.3	1.5
26. I have received threats through devices (internet, phone, mobile, etc.) because of my sexual orientation.	91.8	4.7	2.3	0.8	0.3	0.3	8.2	1.4
27. If someone was ignored because of their gender identity/expression or sexual orientation, I did nothing.	80.7	10.6	5.0	1.6	0.9	1.2	19.3	2.7
28. If someone was threatened because of their gender identity/expression or sexual orientation, I did nothing.	81.4	9.9	5.0	1.9	0.8	1.0	18.6	3.7
29. If someone was hit because of their gender identity/expression or sexual orientation, I did nothing.	79.4	10.2	6.1	1.4	1.4	1.4	20.6	4.2
30. I have spread a rumor about another classmate’s gender identity/expression or sexual orientation.	88.2	6.4	3.6	0.8	0.6	0.4	11.8	1.8
31. If someone bothered a classmate because of their gender identity/expression or sexual orientation, I intervened to stop it.	69.1	11.8	8.9	3.2	1.9	4.9	30.9	10
32. If someone laughed at a classmate because of their gender identity/expression or sexual orientation, I tried to stop it.	64.8	12.9	9.3	4.4	3.1	5.4	35.2	12.9
33. I have supported/accompanied a classmate who has been ignored because of their gender identity/expression or sexual orientation.	62.3	12.0	9.3	4.9	3.4	8.2	37.7	16.5

Notes: Global = One or more times (once + sometimes + many times + almost always + always); Severe = Frequent (many times + almost always + always).

**Table 2 healthcare-13-03119-t002:** Victims, perpetrators, and bystanders of homophobic bullying: Frequencies, Percentages, and Pearson’s Chi-Square according to sexual orientation and sex.

	Total (*n* = 1558)		Hetero (*n* = 1453)		Non-Hetero (*n* = 105)		χ^2^ (*p*)	Male (*n* = 776)		Female (*n* = 782)		χ^2^ (*p*)
	Never	Global	Never	Global	Never	Global		Never	Global	Never	Global	
	f (%)	f (%)	f (%)	f (%)	f (%)	f (%)		f (%)	f (%)	f (%)	f (%)	
Victim	362 (23.4)	1184 (76.6)	349 (24.2)	1093 (75.8)	13 (12.5)	91 (87.5)	7.408 (0.006)	181 (23.5)	588 (76.5)	181 (23.3)	596 (76.7%)	0.013 (0.910)
Perpetrator	1.364 (88.2)	182 (11.8)	1.274 (88.3)	168 (11.7)	90 (86.5)	14 (13.5)	0.306 (0.580)	660 (85.8)	109 (14.2)	704 (90.6)	73 (9.4)	8.499 (0.004)
Bystander	774 (48.1)	802 (51.9)	707 (49.0)	735 (51.0)	37 (35.6)	67 (64.4)	7.031 (0.008)	392 (51)	377 (49.0)	352 (45.3)	425 (54.7)	4.982 (0.026)

Notes: Hetero = Heterosexuals, Non-Hetero = Non-heterosexuals; Victim = victim of homophobic bullying, Perpetrator = perpetrator of homophobic bullying, Bystander = bystander of homophobic bullying; Global = One or more times (once + sometimes + many times + almost always + always); f = frequency; % = percentage; χ^2^ = Pearson’s Chi-Square; *p* = significance level.

**Table 3 healthcare-13-03119-t003:** Means, standard deviations, analysis of variance, and effect size (Cohen’s d) according to sexual orientation among victims and perpetrators of homophobic bullying in relation to positive emotional factors and psychopathological symptoms.

Variables	Victims of Homophobic Bullying			Perpetrators of Homophobic Bullying				
	Hetero (*n* = 1093)	Non-Hetero (*n* = 91)	F(1, 1556) *(p)*	d	Hetero (*n* = 168)	Non-Hetero (*n* = 14)	F(1, 1556) *(p)*	d
	M (SD)	M (SD)			M (SD)	M (SD)		
TMMS-24. Emotional Attention	22.88 (8.40)	22.60 (9.44)	0.08 (0.765)	0.03	23.26 (7.81)	21.64 (10.47)	0.52 (0.469)	0.17
TMMS-24. Emotional Clarity	23.51 (8.52)	22.13 (8.81)	2.19 (0.138)	0.15	23.18 (7.63)	20.00 (8.19)	2.21 (0.138)	0.40
TMMS-24. Emotional Repair	25.71 (8.65)	23.63 (9.04)	4.80 (0.029)	0.23	25.35 (8.02)	25.21 (10.29)	0.00 (0.952)	0.01
TMMS-24. Overall Intrapersonal Emotional Intelligence	72.10 (21.88)	68.36 (23.22)	4.42 (0.120)	0.16	71.80 (19.20)	66.86 (25.62)	0.80 (0.370)	0.21
TECA. Perspective Taking	26.79 (4.47)	25.89 (4.86)	3.28 (0.068)	0.19	25.92 (4.70)	24.71 (3.51)	0.88 (0.349)	0.29
TECA. Emotional Comprehension	29.13 (4.55)	28.96 (5.19)	0.11 (0.736)	0.03	28.33 (4.22)	29.64 (3.50)	1.27 (0.260)	0.33
TECA. Empathic Distress	25.63 (4.59)	24.82 (5.57)	2.47 (0.116)	0.15	24.92 (4.38)	24.86 (4.86)	0.00 (0.958)	0.01
TECA. Empathic Joy	29.93 (5.34)	28.67 (6.24)	4.55 (0.033)	0.21	28.44 (5.44)	28.57 (7.01)	0.00 (0.931)	0.02
TECA. Overall Empathy	111.47 (13.21)	108.34 (15.51)	4.57 (0.033)	0.21	107.61 (13.07)	107.79 (13.42)	0.00 (0.962)	0.01
OHQ. Happiness	112.25 (33.59)	101.20 (35.34)	9.01 (0.003)	0.32	108.31 (31.41)	102.00 (36.92)	0.50 (0.478)	0.18
SAS. Fear of Negative Evaluation	20.58 (7.85)	22.77 (8.54)	6.45 (0.011)	0.26	21.57 (7.29)	22.93 (8.14)	0.43 (0.509)	0.17
SAS. Social Avoidance-Distress of Strangers	16.92 (6.10)	17.86 (6.99)	1.95 (0.163)	0.14	17.31 (5.63)	18.79 (6.95)	0.85 (0.358)	0.23
SAS. Social Avoidance-Distress of Acquaintances	9.94 (4.25)	10.53 (4.89)	1.58 (0.209)	0.12	10.59 (4.08)	10.50 (5.38)	0.00 (0.941)	0.01
SAS. General Social Avoidance and Distress	47.43 (16.42)	51.15 (18.01)	4.25 (0.039)	0.21	49.47 (15.45)	52.21 (19.71)	0.38 (0.534)	0.15
SCL-90. Somatization	0.96 (0.86)	1.49 (0.94)	31.03 (0.000)	0.58	1.10 (0.88)	1.56 (1.04)	3.35 (0.069)	0.47
SCL-90. Obsessive-Compulsion	1.26 (0.95)	1.84 (1.10)	30.43 (0.000)	0.56	1.44 (1.34)	0.91 (1.07)	0.16 (0.685)	0.43
SCL-90. Interpersonal Sensitivity	1.01 (0.90)	1.48 (1.02)	22.12 (0.000)	0.48	1.20 (0.88)	1.49 (1.10)	1.34 (0.248)	0.29
SCL-90. Depression	1.08 (0.88)	1.63 (0.96)	31.74 (0.000)	0.59	1.27 (0.84)	1.84 (1.08)	5.52 (0.020)	0.58
SCL-90. Anxiety	1.00 (0.96)	1.52 (1.11)	23.35 (0.000)	0.50	1.23 (0.87)	1.77 (1.21)	4.54 (0.034)	0.51
SCL-90. Hostility	0.97 (0.95)	1.38 (1.13)	14.25 (0.000)	0.39	1.16 (0.93)	1.80 (1.34)	5.66 (0.018)	0.55
SCL-90. Phobic Anxiety	0.86 (0.91)	1.11 (1.00)	6.19 (0.014)	0.26	1.13 (0.90)	1.55 (1.27)	2.59 (0.111)	0.38
SCL-90. Paranoid Ideation	0.91 (0.88)	1.29 (1.09)	15.12 (0.000)	0.38	1.11 (0.86)	1.71 (1.27)	5.71 (0.018)	0.55
SCL-90. Psychoticism	0.88 (0.86)	1.43 (1.07)	32.26 (0.000)	0.56	1.14 (0.85)	1.75 (1.24)	6.07 (0.015)	0.57
SCL-90. GSI. Global Severity Index	1.00 (0.75)	1.49 (0.87)	34.16 (0.000)	0.60	1.21 (0.72)	1.64 (1.05)	4.30 (0.040)	0.47

Notes: Hetero = Heterosexuals, Non-Hetero = Non-heterosexuals; M = Mean, SD = Standard Deviation, F = Fisher’s F, *p* = significance level; d = Cohen’s d.

**Table 4 healthcare-13-03119-t004:** Frequency and percentages of victims and perpetrators of homophobic bullying who have sought psychological assistance.

	Victims of Homophobic Bullying			Perpetrators of Homophobic Bullying		
	Non-Victim (*n* = 360, 23.4%)	Victim (*n* =1164, 76.6%)	χ^2^ (*p*)	Non-Perpetrator (*n* = 1343, 88.2%)	Perpetrator (*n* = 181, 11.8%)	χ^2^ (*p*)
Psychological assistance	73 (19.2%)	308 (80.8%)	8.19 (0.017)	323 (84.8%)	58 (15.2%)	6.60 (0.037)
No psychological assistance	287 (25.1%)	856 (74.9%)		1020 (89.2%)	123 (10.8%)	

Notes: χ^2^ = Pearson’s Chi-Square; *p* = significance level.

**Table 5 healthcare-13-03119-t005:** Hierarchical Multiple Regression Predicting Homophobic Bullying Victimization.

Step	Predictors Entered	R	R^2^	ΔR^2^	F Change	df1, df2	*p*	Predictor	B	SE B	β	t	*p*
1	Sex, Age, SO	0.155	0.024	0.024	12.40	3, 1503	<0.001						
2	EI, EM, HA	0.228	0.052	0.028	5.51	8, 1495	<0.001						
3	SA, PS	0.431	0.186	0.134	20.31	12, 1483	<0.001						
								Sex	−1.830	0.612	−0.077	−2.987	0.003
								SO	4.052	1.124	0.087	3.606	<0.001
								HA	−0.027	0.010	−0.080	−2.650	0.008
								FNE	0.151	0.060	0.103	2.651	0.008
								SOM	1.440	0.510	0.106	2.825	0.005
								PI	1.580	0.593	0.119	2.654	0.008

Notes: SO = Sexual orientation; EI = Emotional intelligence; EM = empathy; HA = Happiness; SA = Social anxiety; PS = psychopathological symptoms FNE = Fear of negative evaluation; SOM = Somatization; PI = Paranoid ideation. B = unstandardized coefficient; SE = standard error of B; β = standardized coefficient; *p*-values two-tailed. Model summary: Model 1: R = 0.155; R^2^ = 0.024; Adjusted R^2^ = 0.022; F(3, 1503) = 12.398, *p* < 0.001. Model 2: R = 0.228; R^2^ = 0.052; ΔR^2^ (vs Model 1) = 0.028; Adjusted R^2^ = 0.045; F(11, 1495) = 7.466, *p* < 0.001; ΔF = 5.505, Δdf = (8,1495), *p* < 0.001. Model 3: R = 0.431; R^2^ = 0.186; ΔR^2^ (vs Model 2) = 0.134; Adjusted R^2^ = 0.173; F(23, 1483) = 14.721, *p* < 0.001; ΔF = 20.310, Δdf = (12,1483), *p* < 0.001.

**Table 6 healthcare-13-03119-t006:** Hierarchical Multiple Regression Predicting Homophobic Bullying Perpetration.

Step	Predictors Entered	R	R^2^	ΔR^2^	F Change	df1, df2	*p*	Predictor	B	SE B	β	t	*p*
1	Sex, Age, SO	0.084	0.007	0.007	3.56	3, 1503	0.014						
2	EI, EM, HA	0.148	0.022	0.015	2.83	8, 1495	0.004						
3	SA, PS	0.226	0.051	0.029	3.80	12, 1483	<0.001						
								Sex	−0.105	0.036	−0.081	−2.900	0.004
								PA	0.063	0.030	0.090	2.119	0.034

Notes: Notes: SO = Sexual orientation; EI = Emotional intelligence; EM = empathy; HA = Happiness; SA = Social anxiety; PS = psychopathological symptoms; PA = Phobic anxiety. B = unstandardized coefficient; SE = standard error of B; β = standardized coefficient; *p*-values two-tailed. Model summary: Model 1: R = 0.084; R^2^ = 0.007; Adjusted R^2^ = 0.005; F(3, 1503) = 3.560, *p* = 0.014. Model 2: R = 0.148; R^2^ = 0.022; ΔR^2^ = 0.015; Adjusted R^2^ = 0.015; F(11, 1495) = 3.041, *p* < 0.001. Model 3: R = 0.226; R^2^ = 0.051; ΔR^2^ = 0.029; Adjusted R^2^ = 0.036; F(23, 1483) = 3.467, *p* < 0.001.

## Data Availability

The data analyses for this macrostudy are not complete. Therefore, the direct scores on which the conclusions of this article are based are currently not openly available. Once all analyses have been completed and the complementary articles to be conducted with this database have been written, all data will be made publicly available to researchers who wish to analyze them.

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
