# Peer review of "Homophobic Bullying Among Adolescents: Prevalence, Associations with Emotional Factors, Psychopathological Symptoms, and Predictors"

_healthcare, 2025, doi:10.3390/healthcare13233119_

Round 1

Reviewer 1 Report

Comments and Suggestions for Authors

Table 4 presents structural inconsistencies that hinder its interpretation. It is necessary to ensure the consistency of the percentages in both the rows and columns.
The effect size or odds ratio is missing from Table 4

Using stepwise is not recommended; it is considered obsolete because:
"Stepwise variable selection has been a very popular technique for many
years, but if this procedure had just been proposed as a statistical method, it
would most likely be rejected because it violates every principle of statistical
estimation and hypothesis testing."(see Harrell,2015)
Harrell, F. E. (2015). Regression modeling strategies: With applications to linear models, logistic and ordinal regression, and survival analysis (2nd ed.). Springer.
A prominent alternative is Hierarchical (or Sequential) Regression, which is among the most widely used techniques today.

Given that the authors used complex multidimensional scales (SCL-90-R, TECA, TMMS), the robust methodological procedure would have been: To perform a Confirmatory Factor Analysis (CFA) for each scale, in order to test whether the theoretical factor structure adequately fits their Bolivian sample (i.e., to perform psychometric validation instead of just assuming that the scale works). After confirming a good model fit, to report the Omega coefficient or Composite Reliability (CR) directly from the CFA results. The fact that the authors report such high Alphas is, paradoxically, a red flag that could mask problems of multidimensionality or item redundancy.

The article does not report the verification of any basic assumptions of MANOVA, nor does it present the correlation matrix of the dependent variables (DVs).
This omission is critical and undermines the validity of part of the study:
- Omission of the Homogeneity of Covariance Matrices test (Box's M test)
- Absence of Correlation between Dependent Variables
- Lack of Multivariate Normality test

Author Response

1. Table 4 presents structural inconsistencies that hinder its interpretation. It is necessary to ensure the consistency of the percentages in both the rows and columns. The effect size or odds ratio is missing from Table 4

In response to the reviewer’s comments, we have carefully and systematically reviewed Table 4 to ensure the consistency of percentages across both rows and columns. Any discrepancies in the percentages concerning perpetrators have been corrected. In addition, the requested analyses have been conducted, and the relevant information (OR, 95% CI, p) has been incorporated into the text, thereby enhancing the precision and interpretability of the results of the observed differences. We believe these revisions have strengthened the clarity and accuracy of our results (section: results, subsection: psychological help-seeking… page 16, lines 593-603 & Table 4 page 17)

2. Using stepwise is not recommended; it is considered obsolete because: "Stepwise variable selection has been a very popular technique for many years, but if this procedure had just been proposed as a statistical method, it would most likely be rejected because it violates every principle of statistical estimation and hypothesis testing."(see Harrell, 2015). Harrell, F. E. (2015). Regression modeling strategies: With applications to linear models, logistic and ordinal regression, and survival analysis (2nd ed.). Springer. A prominent alternative is Hierarchical (or Sequential) Regression, which is among the most widely used techniques today.

Although we have used the stepwise method in previous studies, and those articles were published in JCR Q1 journals without any criticisms regarding the use of this procedure, following the reviewer’s recommendation, we conducted a hierarchical regression including three sequential blocks (demographics, emotional competence/well-being, and psychopathology/social anxiety). The final model explained 18.6% of the variance in victimization, with significant contributions from sexual orientation, happiness, fear of evaluation, somatization and paranoid ideation. The full set of coefficients, standard errors, β values and significance levels for all three models have been added in a new table (Table 5).

Moreover, we conducted a hierarchical multiple regression to examine predictors of Homophobic Bullying Perpetration (Table 6). The analysis showed that demographic variables explained a small but significant portion of the variance, with sex emerging as a consistent predictor. The inclusion of emotional competence variables produced a modest increase in explained variance, and the final model showed that phobic anxiety was the only psychological variable that significantly predicted perpetration. Overall, the full model accounted for 5.1% of the variance.

The previously included regression information has now been removed (the entire section related to the earlier regression analysis), and two new tables have been constructed with the updated results, along with the corresponding interpretation (section: results, subsection: predictive variables… pages 17-18, lines 612-665 & Tables 5-6).

In addition, the Data Analysis section has been revised to explain and justify the analytical approach (section: data análisis, page 10, lines 446-460).

The results reported in the abstract have also been updated (section: abstract, page 2, lines 51-55).

Further modifications have been made to the Discussion section, including a revised synthesis of findings and confirmation of the study hypothesis (section: discussion, page 20, lines 747-767).

Moreover, the new analysis includes sociodemographic variables, and therefore the formulation of H4 has been rewritten (section: introduction, subsection: objectives and hypotheses, page 6, lines 281-284).

3. Given that the authors used complex multidimensional scales (SCL-90-R, TECA, TMMS), the robust methodological procedure would have been: To perform a Confirmatory Factor Analysis (CFA) for each scale, in order to test whether the theoretical factor structure adequately fits their Bolivian sample (i.e., to perform psychometric validation instead of just assuming that the scale works). After confirming a good model fit, to report the Omega coefficient or Composite Reliability (CR) directly from the CFA results. The fact that the authors report such high Alphas is, paradoxically, a red flag that could mask problems of multidimensionality or item redundancy.

First, we note that the alpha coefficients obtained in the present sample are highly consistent with those reported in the original standardization studies of the instruments, as well as with findings from other Spanish-speaking populations. This consistency suggests that the instruments do not exhibit substantial multidimensionality or item redundancy in this sample.

Second, we fully recognize the importance of conducting psychometric validation of instruments within each specific cultural context. In this regard, the TECA, TMMS-24, SCL-90-R, OHQ, and SAS scales all have previously published validations in Spanish-speaking populations, demonstrating replicated factorial structures and adequate internal consistency. The EBQ was likewise developed using an Argentine sample, providing additional support for its use in Latin American contexts.

We conducted a targeted search for recent cross-cultural validation studies specifically involving Bolivian samples for the instruments used in this study and confirmed that, to date, such studies are not available. Although adaptations and validations have been published for related populations (e.g., TECA in Colombia and Argentina; TMMS in Spain; OHQ-SF with Spanish-speaking samples; EBH in Argentina), we did not identify psychometric evidence based on Bolivian samples for most instruments. With the exception of the TRI version of the SCL-90-R administered in Bolivia (Abal, F.J.P. et al., Ajayu, 17(2), 2019, 424-443), previous research does not encompass Bolivian adolescents. This finding underscores the need for future investigations to conduct confirmatory validations and to report composite reliability indices (e.g., Omega) in Bolivian adolescent populations.

Nevertheless, conducting CFAs for each instrument lies beyond the scope and aims of the present study. However, in accordance with the reviewer’s recommendation, we have added a paragraph to the Limitations section explicitly acknowledging the desirability of future confirmatory validation studies in Bolivian populations, as well as the importance of reporting composite reliability measures such as Omega (section: discussion, page 21, lines 788-791).

4. The article does not report the verification of any basic assumptions of MANOVA, nor does it present the correlation matrix of the dependent variables (DVs). This omission is critical and undermines the validity of part of the study: - Omission of the Homogeneity of Covariance Matrices test (Box's M test) - Absence of Correlation between Dependent Variables - Lack of Multivariate Normality test

We sincerely appreciate this observation. We attempted to compute all assumption tests requested. However, SPSS generated system-level warnings indicating that several tests could not be performed due to intrinsic statistical constraints of the dataset. Specifically:

  • Box’s M test could not be computed because there are fewer than two non-singular cell covariance matrices”. This situation arises when one of the groups presents a singular covariance matrix, which makes matrix inversion impossible. Singular matrices are expected in MANOVA designs involving: (1) a large number of dependent variables that are substantially intercorrelated (e.g., SCL-90-R subscales, emotional intelligence indicators), and (2) highly unbalanced group sizes, as in our case (e.g., heterosexual victims n = 1,093 vs. non-heterosexual victims n = 91; heterosexual perpetrators n = 168 vs. non-heterosexual perpetrators n = 14). These conditions produce near-zero variances and linear dependencies that prevent SPSS from computing Box’s M. This phenomenon is well documented (e.g., Tabachnick & Fidell, 2019; Field, 2018).
  • Multivariate normality tests were not provided because SPSS reported that “no multivariate lack-of-fit tests will be performed because there are too many dependent variables.” When the number of dependent variables is high relative to the residual degrees of freedom, the software cannot generate the multivariate residual matrices required for testing normality. This is a technical limitation of the statistical package, not an omission in our analytic procedure.
  • Regarding the correlation structure among dependent variables: In MANOVA, dependent variables are not required to be uncorrelated; in fact, MANOVA assumes moderate correlations among dependent variables, which is why multivariate analysis is appropriate. The evaluation of zero correlations is therefore not an assumption of MANOVA and would be statistically inappropriate. Moderate intercorrelations between emotional variables and psychopathological symptoms (as expected theoretically) support the use of multivariate analysis.

Despite these computational limitations, the MANOVA remains valid because: (1) MANOVA is robust to moderate violations of multivariate normality, particularly with large sample sizes (Stevens, 2009); and (2) In line with methodological recommendations, we relied on Pillai’s Trace, the most conservative and robust multivariate statistic under violations of assumptions and unequal group sizes (Olson, 1974; Tabachnick & Fidell, 2019).

We have added a paragraph in the results section clarifying these issues and justifying the statistical choices according to best practices in multivariate analysis (section: results, subsection: emotional factors… pages 13-14, lines 530-533).  Although Wilks’ Lambda was initially reported, Pillai’s Trace was used as the primary multivariate test in the present analysis due to technical limitations encountered in SPSS analysis (section: results, subsection: emotional factors… pages 14, lines 536-537 & 562-563). Specifically, Box’s M and multivariate normality tests could not be computed because of singular covariance matrices in the smaller group and the large number of dependent variables. Given its greater robustness to violations of assumptions, including unequal group sizes and non-normality, Pillai’s Trace was considered a more appropriate statistic for interpreting the multivariate results.

Reviewer 2 Report

Comments and Suggestions for Authors
  1. The highlights (Page 1) and abstract (Page 1-2) provide prevalence rates 76.6% victims, 11.8% perpetrators and comparisons 87.5% non-heterosexual vs. 75.8% heterosexual victims but no mention of confidence intervals, standard errors, or p-values to indicate statistical significance.
  2. The highlights (Page 1) state "no gender differences were observed" among victims yet the table (Page 1) notes "a higher proportion of male perpetrators" (11.8%, mostly male) and later text (Page 2) mentions higher male perpetration by heterosexual cisgender peers (Page 4). This inconsistency raises questions about whether gender effects were fully explored or reported.

  3. The sample consists of 1,558 Bolivian students aged 13-17 (Page 1, Abstract) which is a specific demographic. The discussion (Page 6) acknowledges a geographical concentration of studies in Western Europe and North America with limited Latin American representation but does not address potential cultural biases or how Bolivia’s context might skew results compared to global trends  UNESCO’s 42% global rate, Page 3.

  4. Include effect sizes or adjusted mean differences non-heterosexual victims scored 0.5 SD higher on depression, p < 0.01 to provide a measurable context for the reported psychopathology, aiding interpretation of clinical relevance.

  5. the limitations of self-reported data and if possible cross-validate with peer or teacher reports in future studies to improve reliability. Acknowledge this limitation explicitly to temper conclusions.

  6. Include the selection of predictor variables in the methods section and report the adjusted R square or model fit statistics R² = 0.35, p < 0.01 to demonstrate the explanatory power of the regression models.

Best of luck

Author Response

  1. The highlights (Page 1) and abstract (Page 1-2) provide prevalence rates 76.6% victims, 11.8% perpetrators and comparisons 87.5% non-heterosexual vs. 75.8% heterosexual victims but no mention of confidence intervals, standard errors, or p-values to indicate statistical significance.

Following the reviewer’s suggestion, the abstract now includes the relevant statistics (Pearson’s chi-square and p-values) that support our statements regarding victims, perpetrators, and bystanders when comparing these groups by sexual orientation and gender (Table 2) (section: abstract, page 1, lines 34-39). For the sake of consistency, effect size information has also been incorporated into the analyses of variance (Table 3), and the logistic regression analyses (Table 4) (section: abstract, page 2, lines 48-49). To avoid redundancy, this information was not added to the highlights, as it is already reported in the abstract

  1. The highlights (Page 1) state "no gender differences were observed" among victims yet the table (Page 1) notes "a higher proportion of male perpetrators" (11.8%, mostly male) and later text (Page 2) mentions higher male perpetration by heterosexual cisgender peers (Page 4). This inconsistency raises questions about whether gender effects were fully explored or reported.

We thank the reviewer for pointing out the need to include this information. We would like to clarify that the apparent discrepancy arises from the distinction between our study’s findings and previous literature. In our sample, no significant gender differences were observed among victims of homophobic bullying (χ² = 0.013, p > .05), whereas a higher proportion of male students engaged in perpetration (χ² = 8.49, p < .01). The references cited on page 4 summarize findings from other studies, which consistently report that girls and non-heterosexual/LGBTQ+ youth are more often victims. The revised manuscript explicitly acknowledges this difference in the discussion section to clarify that our results do not replicate the gender patterns reported in previous research (section: discussion, page 19, lines 685-694). Therefore, there is no inconsistency in the data; rather, our findings provide a context-specific perspective that differs from prior studies.

  1. The sample consists of 1,558 Bolivian students aged 13-17 (Page 1, Abstract) which is a specific demographic. The discussion (Page 6) acknowledges a geographical concentration of studies in Western Europe and North America with limited Latin American representation but does not address potential cultural biases or how Bolivia’s context might skew results compared to global trends  UNESCO’s 42% global rate, Page 3.

We thank the reviewer for highlighting the importance of addressing cultural context in interpreting our findings. In response, we have expanded the discussion to provide a more detailed account of Bolivia’s specific sociocultural context. While our study addresses the need for research in Latin American populations, we now explicitly acknowledge that Bolivia constitutes a paradoxical context: a progressive legal framework protecting sexual orientation and gender identity coexists with conservative social attitudes, low levels of LGBTI acceptance, limited contact with LGB individuals, and rigid constructions of hegemonic masculinity. These factors, combined with restricted implementation of comprehensive sexuality education and high levels of gender-based violence, may contribute to the higher prevalence of homophobic bullying observed in our sample (76.6%) compared to UNESCO’s global rate (42%). By including this contextual information, we aim to clarify that the results reflect a specific cultural and national context, which may not be directly generalizable to other countries, particularly those in Western Europe or North America, where most studies are concentrated. This addition addresses potential cultural biases and situates our findings within a clear sociocultural framework. The writing of this paragraph has required the inclusion of seven new references (section: discussion, page 19, lines 695-717; and section references, pages 28-29, lines: 1102-1124).

  1. Include effect sizes or adjusted mean differences non-heterosexual victims scored 0.5 SD higher on depression, p < 0.01 to provide a measurable context for the reported psychopathology, aiding interpretation of clinical relevance.

We thank the reviewer for this insightful and valuable comment. Following the recommendation, we have now incorporated a detailed interpretation of effect sizes (Cohen’s d) in the Results section for both victims and perpetrators of homophobic bullying. Specifically, we added text explaining the magnitude and clinical relevance of the differences observed between heterosexual and non-heterosexual participants across psychopathological and emotional variables. This new content clarifies the practical significance of the findings and enhances readers’ ability to understand the psychological impact associated with each profile. The added interpretation appears in the Results section (section: results, page 14, subsection: emotional factors, lines 546-559 & 570-578) and directly addresses the reviewer’s concern by providing meaningful context beyond statistical significance. We appreciate the reviewer’s suggestion, which has undoubtedly strengthened the clarity and interpretability of the manuscript.

  1. The limitations of self-reported data and if possible cross-validate with peer or teacher reports in future studies to improve reliability. Acknowledge this limitation explicitly to temper conclusions.

Taking this suggestion into account, we now acknowledge social desirability bias as a limitation inherent to self-reported data, and we recommend cross-validation with peer or teacher reports in future studies to enhance reliability (section: discussion, page 21, lines 783-784).

  1. Include the selection of predictor variables in the methods section and report the adjusted R square or model fit statistics R² = 0.35, p < 0.01 to demonstrate the explanatory power of the regression models.

Thank you very much for highlighting the relevant information that needed to be included. The other reviewer suggested replacing the stepwise linear regression analyses originally conducted with hierarchical (sequential) regression analyses, as this approach was deemed more appropriate. Accordingly, to identify the predictor variables of homophobic bullying (victimization and perpetration), we now report hierarchical (sequential) regression analyses.

In the Methods section, we have added detailed information regarding the selection of predictor variables, as well as the rationale for their inclusion (section: materials & methods, subsection: data analysis, page 10, lines 446-460).

The Results section now incorporates new tables and revised interpretations of the findings, including all coefficients requested (R, R², p, etc.) to demonstrate the explanatory power of the regression models (section: results, subsection: predictive variables… pages 17-18, lines 612-665 & Tables 5-6).

These results are further discussed in depth in the Discussion section, where Hypothesis 4 is also addressed (section: discussion, page 20, lines 747-767).

In addition, the new analysis includes sociodemographic variables, and therefore the formulation of H4 has been rewritten (section: introduction, subsection: objectives and hypotheses, page 6, lines 281-284).

Round 2

Reviewer 1 Report

Comments and Suggestions for Authors

With the modifications, it is ready to be published.